# Transgressions of compartment boundaries and cell reprogramming during regeneration in *Drosophila*

**Salvador C Herrera, Ginés Morata\***

Centro de Biología Molecular Severo Ochoa, Universidad Autónoma de Madrid, Madrid, Spain

**Abstract** Animals have developed mechanisms to reconstruct lost or damaged tissues. To regenerate those tissues the cells implicated have to undergo developmental reprogramming. The imaginal discs of *Drosophila* are subdivided into distinct compartments, which derive from different genetic programs. This feature makes them a convenient system to study reprogramming during regeneration. We find that massive damage inflicted to the posterior or the dorsal compartment of the wing disc causes a transient breakdown of compartment boundaries, which are quickly reconstructed. The cells involved in the reconstruction often modify their original identity, visualized by changes in the expression of developmental genes like *engrailed* or *cubitus interruptus*. This reprogramming is mediated by up regulation of the JNK pathway and transient debilitation of the epigenetic control mechanism. Our results also show that the local developmental context plays a role in the acquisition of new cell identities: cells expressing *engrailed* induce *engrailed* expression in neighbor cells.

**\*For correspondence:**
gmorata@cbm.uam.es

**Competing interests:** The authors declare that no competing interests exist.

## Introduction

Regeneration is a classical problem in developmental biology. The cells involved have to undergo a developmental reprogramming so that they acquire the new cell identities necessary to regenerate a lost organ or a damaged tissue. In the classical example of the regeneration of an amputated amphibian limb (*Kragl et al., 2009*), proximal cells have to generate new cells with different—distal—identity. Very little is known about the phenomena and mechanisms behind this developmental reprogramming. We are investigating this problem using the compartments of *Drosophila* imaginal discs as a model system for regeneration.

The subdivision into lineage blocks, termed compartments, is a principal feature of the organization of the body of *Drosophila* (*Garcia-Bellido et al., 1973*; *Morata & Lawrence, 1977*). The earliest compartmentalization event is established during embryogenesis (*Lawrence & Morata, 1977*) and separates anterior (A) and posterior (P) compartments in each segment. The A/P boundaries in the different body segments not only separate the original lineage blocks of the fly; they also are topological landmarks to delimit Hox genes expression (*Lawrence, 1992*).

Since the initial establishment of the A/P boundary, the P compartments acquire expression of the *engrailed (en)* gene, which determines anterior or posterior identity within each segment: the 'on' state specifies posterior whereas the 'off' state specifies anterior identity. This early lineage segregation is preserved for the rest of the development in all larval and adult structures. In the imaginal discs (the precursors of the adult cuticular structures) *en* remains expressed in the P compartments (*Brower, 1986*) and is also required to maintain the A/P border (*Morata and Lawrence, 1975*). The absence of *en* function in the A compartment allows anterior cells to activate the Hh pathway and subsequently the *cubitus interruptus (ci)* gene, a marker of A compartment identity (*Orenic et al., 1990*).

**eLife digest** When cells or tissues are damaged, animals can often regenerate the affected tissues. In an effort to identify the genes and mechanisms that are involved in this regeneration process, researchers often perform experiments on relatively simple organisms or systems. These experiments frequently involve the amputation of specific cells or organs so the researchers can observe and manipulate the events that occur during the subsequent regeneration.

One such model organism is the fruit-fly *Drosophila*. Inside the *Drosophila* larva are structures called imaginal discs, which are precursors to parts of the outer cuticle of the adult fly. Each imaginal disc contains two main boundaries, dividing it into anterior/posterior and dorsal/ventral compartments: posterior cells, for example, express a gene called *engrailed* to produce the relevant protein, whereas anterior cells do not; likewise, the gene *apterous* is expressed in dorsal cells but not ventral cells. These genes, *engrailed* and *apterous*, are the key factors that determine the developmental features–and hence the identity—of the posterior and the dorsal cells respectively.

Herrera and Morata investigated how cells regenerate when parts of the imaginal disc are destroyed, using a genetic technique that causes high levels of programmed cell death in either the posterior or the dorsal compartments of the disc.

Destroying most of the cells in either of these compartments in the imaginal disc leads to a temporary breakdown of the corresponding boundary, which is then rapidly reconstructed. During this reconstruction process, some of the imaginal disc cells are reprogrammed: for example, if the cells in the posterior compartment are destroyed, some anterior cells take on a posterior identity. This reprogramming occurs because the cell destruction disrupts the way that the expression of genes such as *engrailed* and *apterous* is controlled by other genes.

Neighboring cells can also control gene expression, and therefore cell identity. Cells that express *engrailed*, for example, cause their neighbors to express *engrailed* too. This process is likely to involve group interactions that might help the distinct compartments in the imaginal disc to form by ensuring that adjacent cells develop in the same way. Similar processes may also occur as part of the normal development of organisms.

The expression of *en* is regulated epigenetically. It is kept in *off* (silenced) state in the A compartments through the activity of the Polycomb-G genes (Pc-G) (**Busturia and Morata, 1988**), whereas the trithorax-G genes (trx-G) maintain *en* in *on* state in P compartments (**Breen et al., 1995**).

In addition to the A/P boundary, there is in the wing disc another lineage border, separating the dorsal (D) and the ventral (V) compartments. This boundary appears during larval development (**Garcia-Bellido et al., 1973**) and is dependent on the activity of the gene *apterous* (*ap*), which confers dorsal identity (**Diaz-Benjumea and Cohen, 1993**).

The A/P and the D/V borders not only separate cells with different identities in the wing disc; they also function as developmental organizers of the imaginal discs. The posterior compartment cells secrete the Hedgehog (Hh) morphogen that activates the *decapentaplegic (dpp)* gene in the anterior cells close to the border, from where the Dpp signal diffuses to the two compartments and controls their growth and pattern (**Lawrence and Struhl, 1996**). The Wg signal emanating from the D/V border also has a major patterning function in the wing disc (**Irvine and Vogt, 1997**).

It follows that the establishment and maintenance of the A/P and D/V boundaries are critical factors in the development of the wing disc and that major alterations of these boundaries would grossly interfere with normal pattern and growth. Therefore, it is expected that regenerative processes incorporate mechanisms to ensure the restoration of these borders in damaged discs. The possibility of collapse and subsequent restoration of the A/P border during regeneration in the wing disc was suggested by **Szabad et al. (1979)**, although recent reports (**Bergantinos et al., 2010**; **Smith-Bolton et al., 2009**) have failed to observe lineage transgressions after massive damage to the disc.

We are investigating the regenerative response of imaginal discs to the ablation of specific compartments, and especially how the stability of compartment boundaries is maintained. Our results indicate that ablation of the P or the D compartment causes a transient collapse of the A/P or the D/V boundaries, which are very quickly reconstructed. During the reconstruction process some cells are reprogrammed and change their original compartmental identity. We provide evidence

that the identity changes are associated with gain of activity of the JNK pathway and with relaxation of the epigenetic control by the Pc-G and trx-G genes. In addition, we suggest that the identity changes observed are induced by a novel mechanism by which isolated cells acquire the identity of their neighbours.

## Results

### Experimental design

In the experiments to induce massive damage to specific compartments, equivalent to ablation, we have utilized the Gal4/UAS/Gal80^TS method (see 'Materials and methods' for details) to force high levels of apoptosis in specific domains (*Bergantinos et al., 2010*; *Smith-Bolton et al., 2009*; *Herrera et al., 2013*). The temperature-sensitive Gal4 suppressor Gal80 allows temporal manipulation of Gal4 activity. In these experiments a temperature shift from 17 to 29°C causes a loss of Gal80 function, which now permits the activity of the Gal4 driver. In turn, Gal4 activates the UAS-*hid* and UAS-*Flp* transgenes (*Figure 1—figure supplement 1*).

The activation of UAS-*hid* with the *hh-Gal4* driver causes massive apoptosis in the P compartment. In addition, the high levels of Flipase generated by the UAS-*flp* transgene induce recombination in the *act>stop>lacZ* cassette, with the result that the cells of the original P compartment become indelibly marked with the activity of the *lacZ* gene, which encodes the ßGal protein. Under our experimental conditions (48 hr of continuous Flp activity) all the cells of the P compartment acquire the ßGal label (*Figure 1A,B*). We refer to these cells as the 'Hh lineage'. This is an important point because we consider the ßGal label as the indicator of the compartmental provenance of the cells.

To ablate the dorsal compartment and to track the lineage of the dorsal compartment cells, we have followed a similar strategy, but using the *ap-Gal4* line, which is expressed specifically in the dorsal wing compartment.

### Regenerative response during and after ablation of the posterior compartment of the wing disc

We describe in detail the results obtained ablating the P compartment. After 48 hr of *hid* activity, the great majority of P compartment cells in the disc are dead or are dying (*Figure 1B*). We estimate that about 70% of the cells die and as a result the size of the compartment is greatly reduced (compare *Figure 1C,D*). However, in spite of the massive cell death in the P compartment, there is clear, although highly irregular (*Figure 1E–G*), boundary separating anterior and posterior cells, outlined by the anterior compartment marker Ci. After allowing 72 hr of recovery at 17°C, the damaged P compartment is completely regenerated: it exhibits normal size and shape and an almost normal A/P border.

We have studied by clonal analysis the growth of the A and P compartments during and after the ablation period. The size of clones in the A and P compartment during the ablation period is similar (*Figure 2A–D*), although the P compartment clones have probably grown more since some of their cells must have died due to Hid activity. The proliferative response to the ablation becomes evident during the recovery time (*Figure 2E–H*): clones in the P compartment are about 3 times bigger than those in the A compartment. These additional divisions restore the full size of the compartment. This result also illustrates the independent size regulation of the two compartments (*Martin and Morata, 2006*); once the A/P border is re-established the two compartments grow autonomously.

### Transgression of the A/P and D/V compartment boundaries during regeneration

Next, we examined if the ablation of the P or the D compartment affect the stability of the A/P or the D/V border. Our experimental system permits distinguishing the compartmental origin of the cells. Those of the P or D compartment are labeled by *lacZ* (ßgal) activity, whereas the A or V compartment cells are labeled by the lack of it.

In the majority of discs in which the P compartment is ablated (22 out of 31), we find that cells from the A compartment have contributed to the regenerated P compartment. The transgressing cells are of anterior origin, as indicated by the lack of ßGal staining, but now belong to the P compartment, as showed by the lost the Ci anterior marker (*Figure 3A*) and the gain of *en* expression (*Figure 3—figure supplement 1A*). These lineage violations are not observed in control discs. There was the possibility, however, that in the experimental discs the ßGal label of the P compartment might not be as

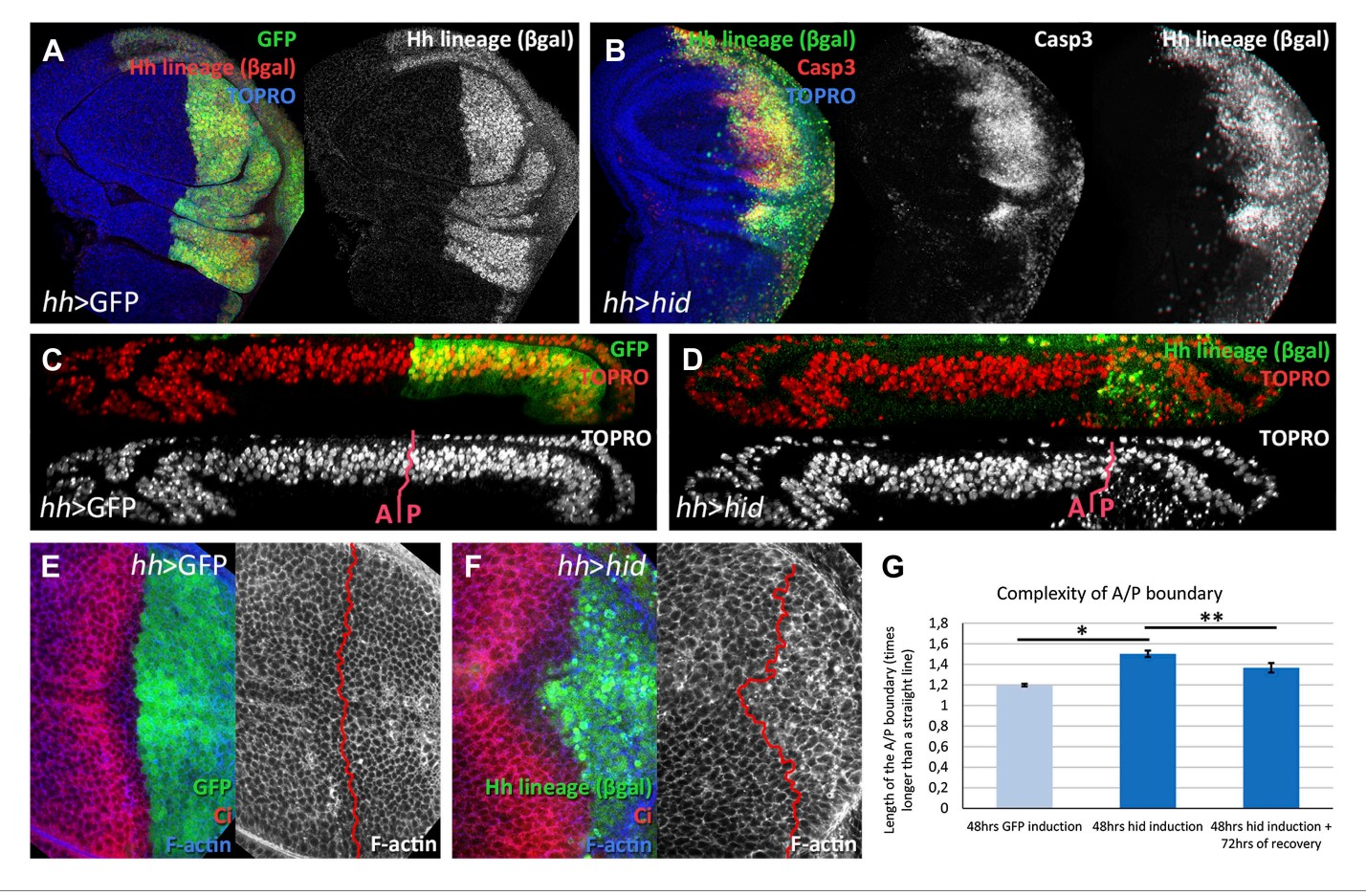

**Figure 1**. Ablation of the posterior compartment. Wing imaginal discs after 48 hr of induction of GFP (**A**, **C** and **E**, controls) or *hid* (**B**, **D** and **F**). Note that in both cases the βgal lineage label is acquired by all the posterior cells (**A** and **B**). The disc in **B** shows high levels of apoptosis, indicated by Caspase3 activity. (**C** and **D**) Cross-sections of wing discs perpendicular to the A/P border at the level of the wing pouch. The peripodial membrane is on top, the columnar epithelium on the bottom. Anterior compartment at the left, posterior at the right. There is a marked reduction of the size of P compartment in which *hid* is expressed (**D**). The epithelium is much thinner than in control disc (**C**), indicating a big reduction of cell number due to the ablation (compare the number of nuclei in the right part of **C** and **D**). (**E**–**G**) Shape of the A/P border after 48 hr of GFP (**E**) or *hid* (**F**) induction. In the later this border becomes wiggly and inter-digitized, a feature quantified in panel **G**. By allowing 72 hr of recovery this effect is partially recovered (third bar in panel **G**). Bars represent S.E.M., n>15 in each genotype and time point, *p<0.01, **p<0.05. See also *Figure 1—figure supplement 1*.

The following figure supplements are available for figure 1:

**Figure supplement 1**. Genetic ablation and lineage labeling system.

precise as in the controls, leaving some cells unmarked that could be interpreted as of anterior origin. But under that hypothesis these unmarked cells should appear anywhere in the P compartment. We find that the transgressions localize to the proximity of the A/P border (*Figure 3—figure supplement 2*), which argues strongly against that possibility.

In addition, we unexpectedly find that surviving cells from the ablated P compartment may also contribute to the A compartment: cells of posterior provenance, labeled with ßGal, frequently penetrate (13 transgressions in a sample of 31 discs) into the A compartment and acquire anterior identity, visualized by *ci* activity (*Figure 3B* and *Figure 3—figure supplement 2*). Since the A compartment was not in need of reconstruction, this observation indicates that the transgressions are caused by a transient collapse of the A/P boundary, which affects the two compartments. It is worth mentioning that the A/P transgressions have occurred during the ablation period, as they are found in discs fixed at the end of it, without recovery time (*Figure 3—figure supplement 1A*, see also *Figure 4E–E''*). This is a significant observation because it suggests that the restoration of the A/P border is immediate and

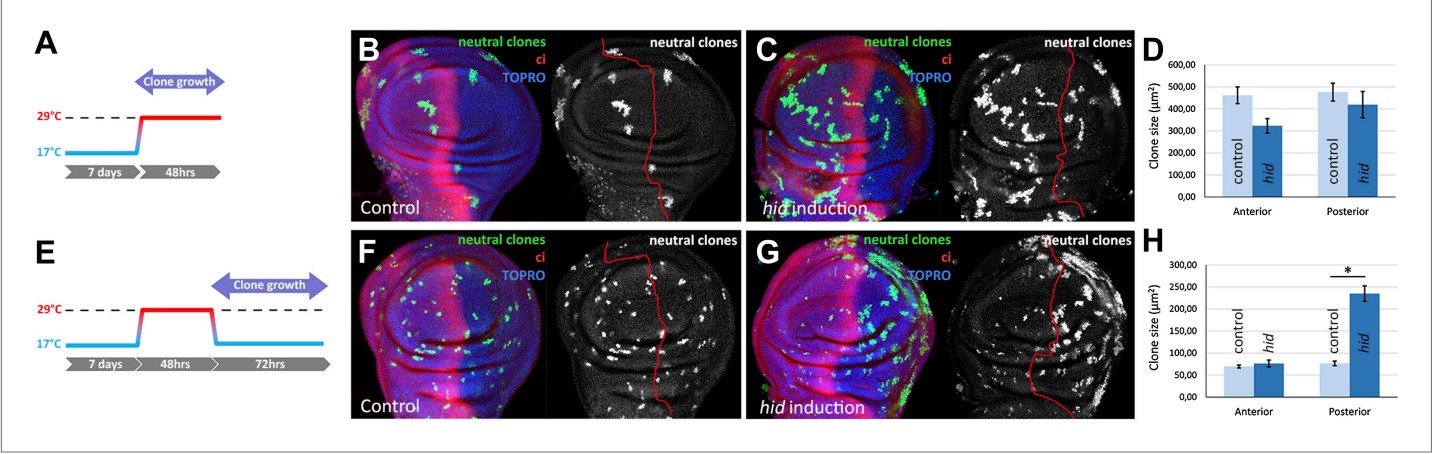

**Figure 2**. Clonal analysis of growth in discs in which the P compartment has been ablated. (**A–D**) GFP-labeled clones only allowed growing during the ablation period. The clones were induced at the beginning of the ablation and the discs fixed at the end of it. Control disc (**B**) a *hid*-induced disc (**C**) and clone size quantification (**D**) are shown. Note the absence of major differences in clone size between control discs and *hid*-induced in both compartments. (**E–H**) Clones induced at the end of the ablation period and scored after 72 hr of recovery period at 17°C (see diagram **E**). Again, a control disc (**F**), a *hid*-induced disc (**G**), and quantification (**H**) are shown. Note in **G** that the clones in the P compartment are much bigger than those in the A compartment of the ablated discs and those of the P compartment in controls (**F**). Bars represent S.E.M., n>25 in each genotype and time point, *p<0.001.

concomitant with the ablation. The lineage violations are not specific to the UAS-*hid* transgene; we have made similar observations using other pro-apoptotic genes (***Figure 3—figure supplement 1B***).

To reinforce the conclusion for the preceding experiments that the A/P border is transgressed during regeneration of the P compartment, we also performed an experiment of clonal analysis. The idea was that if the A/P border had collapsed during ablation some marked clones induced before or at the time of collapse should include anterior and posterior cells. We generated 'twin clones' such that we labeled the progeny of the two cells resulting from mitotic recombination events (see 'Materials and methods'). The clones were generated at the beginning of the ablation period and fixed the discs at different times after the end. As illustrated in ***Figure 3C***, we found numerous cases of clones crossing over the A/P border (10 cases of crossing in 16 discs examined). These transgressions do not appear in discs of the same genotype in which the P compartment is not ablated.

A loss of lineage restriction was also observed in the D/V compartment boundary after ablation of the D compartment. As in the experiment to assay the A/P boundary, we can follow the lineage of the D compartments by labeling the cells with the UAS-Flipase system (see 'Materials and methods' and ***Figure 1—figure supplement 1***). We find that cells of ventral provenance appear integrated in the regenerated D compartment (***Figure 3D***). The change of identity is visualized by the acquisition by these cells of PS1α integrin, a marker of dorsal cells (***Gotwals et al., 1994***).

## JNK pathway activity is required for the A/P boundary transgressions

There is evidence that JNK signaling is involved in *in situ* regeneration in imaginal discs (***Bergantiños et al., 2010***; ***Herrera et al., 2013***; ***Worley et al., 2012***) and also in transdetermination of transplanted discs (***Lee et al., 2005***). Moreover, a recent report (***Gettings et al., 2010***) indicates that JNK activity is required for the change of *en* expression and subsequent crossing of the A/P border by some cells during dorsal closure in *Drosophila* embryos. Therefore, we have examined whether it plays a role in the lineage transgressions we observe. We had previously noticed that JNK is up regulated in the P compartments of ablated *hh > hid* discs, as shown by the activation of the *puc*-LacZ marker, which is not normally expressed in the wing pouch (***Figure 4B,B'***). This was expected because Hid induces JNK (***Shlevkov and Morata, 2012***), but we also find anterior compartment cells near the A/P border that express the *puc*-LacZ marker (***Figure 4A–A''***).

We designed a special experiment combining the lexA/lexO (***Yagi et al., 2010***) and the Gal4/UAS binary systems to examine the involvement of JNK in the A/P lineage transgressions. The rationale was to induce massive cell death in the region of the A compartment close to the A/P border, and at the same time to prevent activation of JNK in the P compartment (***Figure 4C***, details in 'Materials and

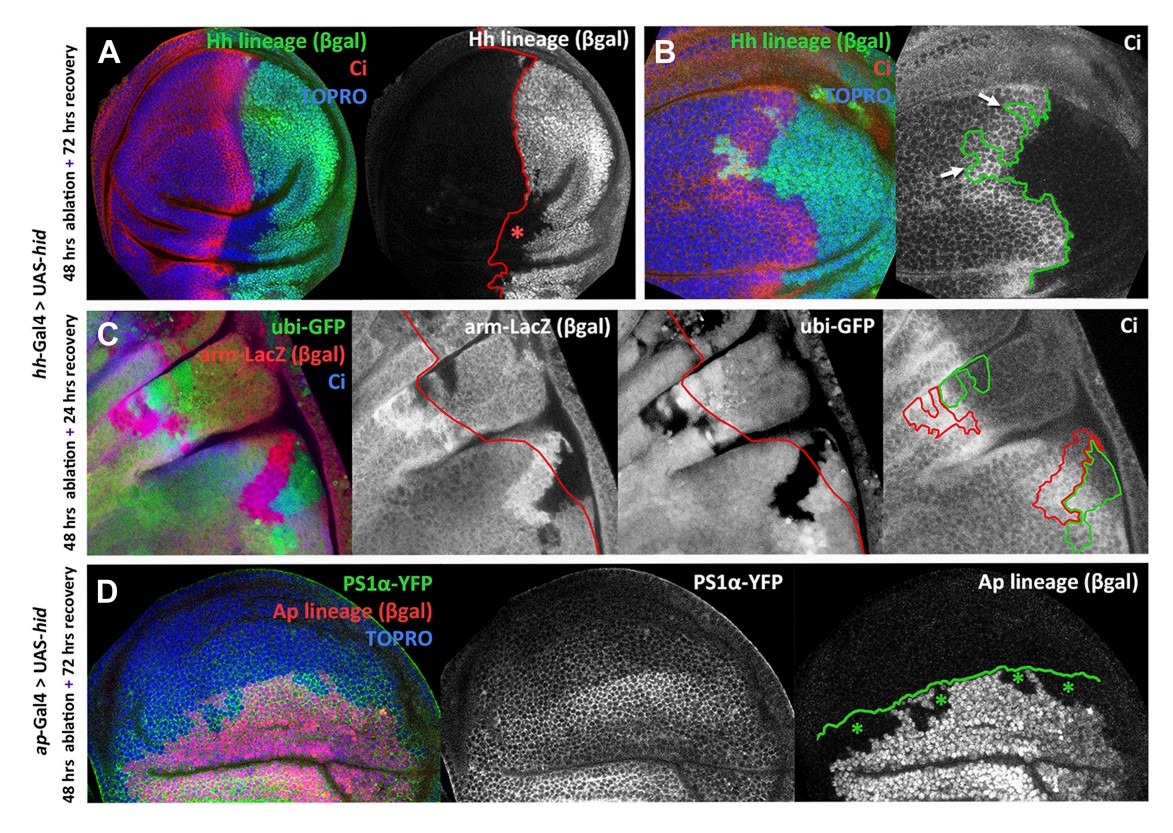

**Figure 3**. Transgressions of the A/P and the D/V border caused by ablation of the P or the D compartment. (**A** and **B**) Wing discs after 48 hr of Hid treatment followed by 72 hr of recovery. The original Hh lineage is labeled by ßgal (green), the A compartment is marked with an anti-Ci antibody (red) and the nuclei with Topro (blue). (**A**) Note the presence of groups of cells (asterisk) originated in the anterior compartment (they lack the βgal label) but that have lost Ci activity, indicating that they have lost anterior identity. (**B**) The unexpected finding that cells from the P compartment—they are part of the Hh lineage—can penetrate in the A compartment and to acquire anterior identity, as demonstrated by the Ci marker (arrows) (**C**) Portion of a disc in which the P compartment has been ablated, containing two sets of 'twin' clones labeled with GFP-green/LacZ ßgal (see 'Materials and methods' for details). The clones were initiated at the beginning of the ablation of the P compartment and fixed 24 hr after the end of ablation. In those panels the A/P borderline is blue and the twin clones delineated in red or green. Note that the two cases the clones cross over the A/P line. (**D**) Wing disc after 48 hr of Hid induction in the dorsal compartment followed by 72 additional hours of recovery. Note the presence of groups of cells (asterisks) that in spite of their ventral origin (lack of dorsal lineage βgal label) now present dorsal markers as the *mew* Integrin (PS1α). See also *Figure 3—figure supplements 1 and 2*.

The following figure supplements are available for figure 3:

**Figure supplement 1**. Transgressions detected during the ablation period and transgressions using *p53* as apoptotic inducer.

**Figure supplement 2**. Localization on the wing disc of the different transgressions of the A/P border.

methods'). In discs of genotype *dpp*-LHG>LexO-*hid hh*-Gal4>UAS-*puc* Gal80^TS the 17 to 29°C temperature shift causes ablation of the Dpp domain in the A compartment and at the same time prevents JNK activation in the P compartment due to the over-expression of the negative regulator *puc* (**Martin-Blanco et al., 1998**). In control *dpp*-LHG>LexO-*hid* Gal80^TS discs the shift causes ablation of the Dpp domain but JNK activity is not prevented. The results are illustrated in *Figure 4D–F*. In the experimental discs, the transgressions are greatly reduced, clearly indicating a requirement for JNK function. This experiment also shows that A/P transgressions also occur after damage to the A compartment, even if only a portion of the compartment is affected.

## Alteration of the Pc-G and trx-G activities during ablation

The preceding experiments establish that after massive damage to the A, P or the D compartment the lineage boundaries collapse but are quickly restored. During the process some cells change their

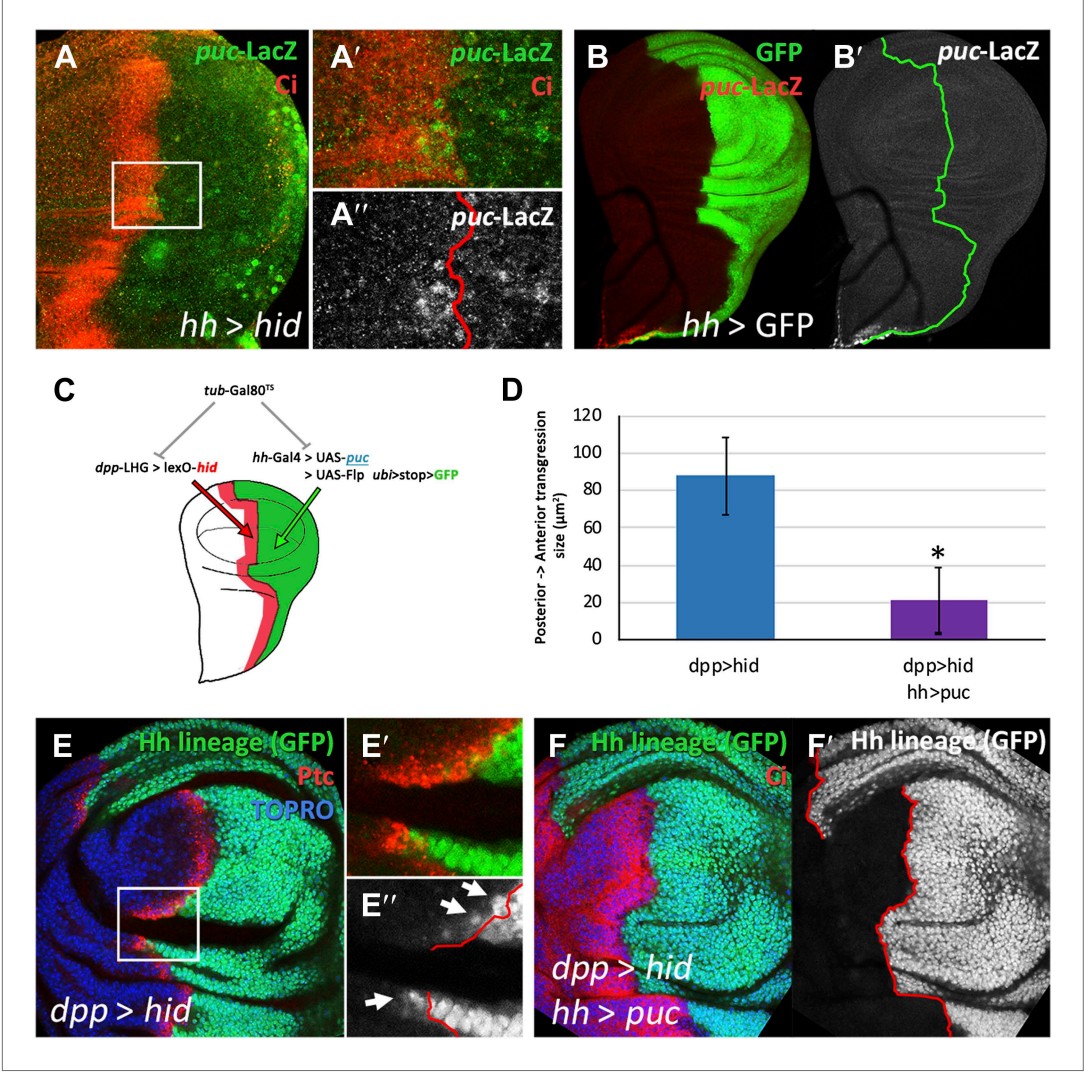

**Figure 4**. Involvement of the JNK pathway in the transgression of the A/P boundary during regeneration.
(**A–A''**) Wing disc and magnification showing activation of *puc*-LacZ (green) after ablation of the P compartment.
The A/P boundary is outlined by Ci (red) label. Note that some anterior cells contain *LacZ* expression. The puc–lacZ
staining is not particularly strong in the P compartment because it is located predominantly on the basal side of the
disc, where the dying cells accumulate. (**B** and **B'**) Control non-ablated disc showing absence of *puc-LacZ* activity,
except in the stalk region where it is normally expressed. The panel **C** illustrates the experiment designed to test
the requirement for JNK activity for crossing the A/P boundary during regeneration. After removing repression by
Gal80$^{TS}$ the *dpp*-LHG line forces *hid* activity in the *dpp* domain (red), located just anterior to the A/P line. At the
same time the posterior compartment cells lose the potential to gain JNK activity due to *puc* over-expression. The
results are illustrated in **D–F'**. Cells of posterior origin (green) can penetrate in the A compartment if they can
activate JNK (**E–E''**), but are unable to do so if JNK activation is prevented by *puc* over-expression. Quantitative
data are shown in **D**. Bars represent S.E.M., n = 15 in each genotype, *p<0.05.

original identity, indicating that those cells have undergone a reprogramming process that modifies
the expression of identity genes, *en, ci, ap*, etc, activating some and repressing others. In support
of this, we observe that after ablation of the dorsal compartment (in *ap>hid* discs, ***Figure 5—figure
supplement 2***) the expression of *engrailed* in the dorsal part of the posterior compartment is lower
than in the corresponding region in the ventral compartment.

The functional state of *Drosophila* identity genes, i.e., their active or silencing mode, is control-
led by the genes of the *Polycomb* (*Pc-G*) or *trithorax* groups (*trx-G*) (***Kennison and Tamkun, 1988***;
***Lewis, 1978***). The Pc-G genes are responsible for keeping in *off* state (silencing) the genes

inappropriate for a particular identity, whereas the trx-G genes maintain the activity of identity-specific genes. Therefore, we checked whether the identity changes observed during the reconstruction of compartment boundaries are associated with alterations of the activity of the Pc-G and the trx-G genes. It is known that the Pc-G and trx-G genes control *en* expression in the A and P compartments (*Busturia and Morata, 1988*; *Breen et al., 1995*).

We have measured the boundary transgressions after damage to P compartments of discs containing only one dose of *Polycomb* (*Pc³/+*) or *trithorax* (*trx^E2/+*) in comparison with controls containing normal doses of the genes. The results indicate an involvement of the Pc-G and trx-G genes (*Figure 5A–G*): in *Pc³/+* discs, the number and size of lineage transgressions is greatly increased with respect to controls, whereas in *trx^E2/+* discs there are very few lineage transgressions.

We have screened the expression during regeneration of several Pc-G and trx-G genes and of some epigenetic markers (*Figure 5—figure supplement 1*). The QF line ET40 (*Potter et al., 2010*) is an insertion of the QF transcription factor in the *Posterior sex combs* (*Psc*) gene, a member of the Pc-G. In *Psc*-QF>QUAS-mtdTomato discs in which the P compartment is ablated there is a clear down regulation of *Psc* activity (*Figure 5—figure supplement 1A,B*). We also find an increase in the levels of tri-methylation of H3K4, a mark of active chromatin (*Schubeler et al., 2004*) (*Figure 5—figure supplement 1C,D*). In both cases, the modifications affect the entire posterior compartment and the region of the A compartment close to the A/P border.

Taking everything together the preceding results strongly suggest that the changes of identity during regeneration are associated with temporal relaxation of the epigenetic control of compartment identity. We conjecture that as a result cells in the neighborhood of the A/P (or D/V) boundary may lose some of their developmental constraints, thus descending to a kind of 'naive' (undetermined) state.

## A mechanism of *en* induction by neighbor cells

During the reconstruction of the boundaries some cells acquire a new compartmental identity, different from the original one. This implies *the novo* activation of identity genes. We speculated about the possibility of an induction mechanism by neighbor cells. That is, a cell in a naive state could be induced by its neighbors to express the same identity gene. The reason behind this speculation was the realization that identity changes observed appear to be influenced by the local developmental context, that is, the cells may change from anterior to posterior, but keep the overall wing identity. This suggested that the local developmental environment could play a role in determining the acquisition of the new identity. There is a recent observation (*Garaulet et al., 2008*) that dorsal compartment cells expressing *en* appear to induce the *en-LacZ* reporter in ventral cells located in their vicinity.

We designed two experiments to check whether A compartment cells might be induced to express the posterior identity gene *en* by the influence of neighbor cells expressing the gene. The first experiment consisted of filling the anterior wing compartment almost entirely with GFP-marked clones of cells expressing *en*, but leaving a few islands of unmarked anterior cells without exogenous *en* activity. We find that these unmarked cells often activate the endogenous *en* gene, as indicated by the expression of the *en-lacZ* insert (*Figure 6C–E''*) and also by the presence of the En protein (*Figure 6—figure supplement 1A–B''*). The gain of *en* activity is associated with down regulation of the A compartment markers Ci and Ptc (*Figure 6D'',E''*, *Figure 6—figure supplement 1C–D''*). Significantly, this activation only occurs in discs densely populated by *en*-expressing clones; it does not appear if there are few clones (*Figure 6A–B''*). It suggests that the induction process requires the receiving cells to be largely surrounded by inducing cells. This experiment was also performed at 17°C, a temperature at which the Gal4 system is less active and therefore the levels of *en* over-expression in the clones are lower. The result (*Figure 6—figure supplement 2*) is that *en* is also non-autonomously activated under these conditions.

The second experiment consisted of generating clones of cells that are refractory to Gal4 in a field of Gal4-driven *en*-expressing cells (see 'Materials and methods'). Clones in the anterior compartment expressing the Gal4-suppressor Gal80 are surrounded by cells with high levels of *en* activity driven by the *act*-Gal4 line. In these clones, we also observe induction of endogenous *en* activity (*Figure 6F–G'*).

We believe that the phenomenon of *en* induction it is not due to the Hedgehog -mediated late activation of *en* in the anterior cells close to the A/P border (*Blair, 1992*), because the *en*-lacZ insert line we use in these experiments does not show this effect (*Figure 6H,H'*). Furthermore, as noted above, we only observe *en* induction in discs densely populated by *en*-expressing clones; in discs with fewer clones it does not occur (compare A–B'' and C–D'' panels in *Figure 6*). The Hh-mediated activation would not depend on clone density.

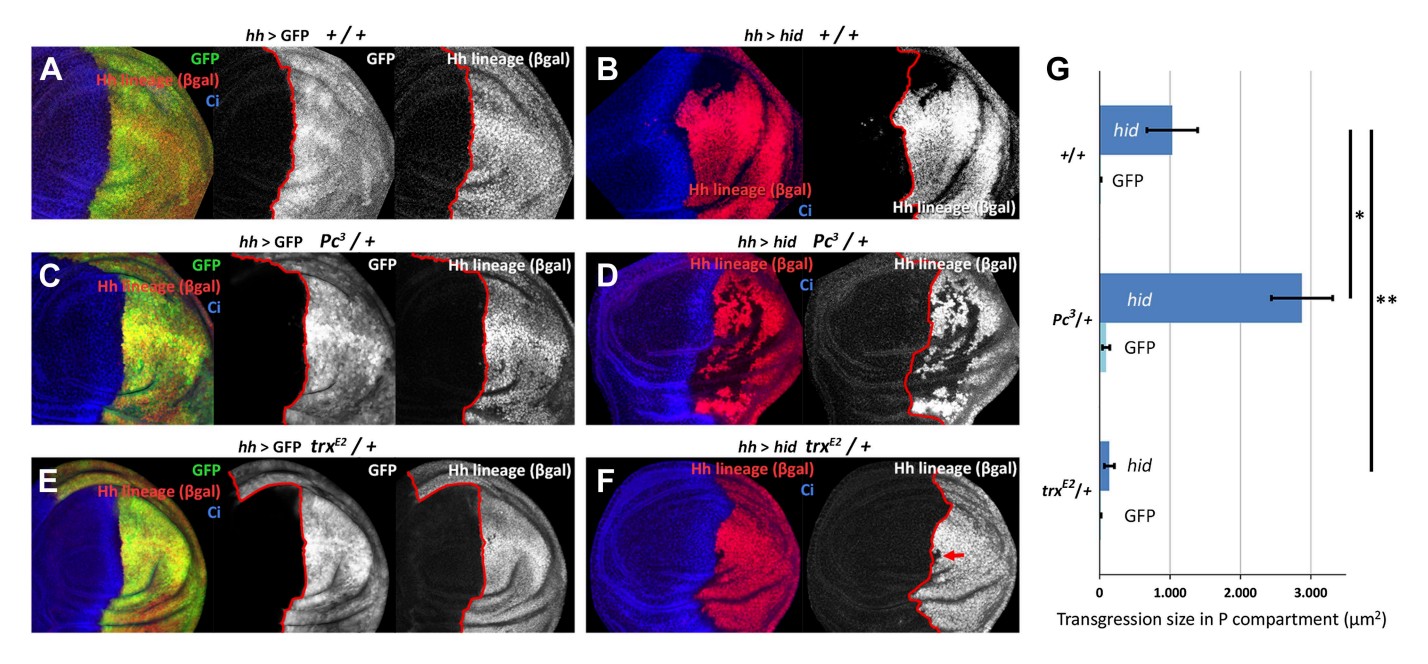

**Figure 5**. Changes in epigenetic regulation during A/P boundary reconstruction. Comparison of the size of transgressions after Hid treatment found in discs containing normal doses of *Polycomb (Pc)* or of *trithorax (trx)* (**A**, control with GFP treatment, **B** experimental), with those in discs containing one dose of *Pc* (**C**, control, **D** experimental) or of *trx* (**E**, control, **F** experimental). Note the significant increase in transgressions size in *Pc³/+* background and the reduction in *trxᴱ²/+* background (a small transgression is indicated with an arrow in **F**). Panel **G** shows the quantification of the transgressions size in the three genotypes, each with respect to its own control. Bars represent S.E.M., n>15 in each genotype, *p<0.01, **p<0.05. See also *Figure 5—figure supplements 1 and 2*.

The following figure supplements are available for figure 5:

**Figure supplement 1**. Alterations in the expression of *Posterior sex comb (psc)* and of H3K4 levels in the proximity of the A/P border after Hid administration.

**Figure supplement 2**. Engrailed protein levels are reduced in posterior cells.

## Discussion

The results reported bear on the regenerative response of the wing disc to ablation of the posterior or the dorsal compartment. Compartments are independent lineage blocks and express different genetic/developmental programs (*Lawrence, 1992*). Since the majority of our experiments concern the response of the P compartment, we restrict the discussion to those, although we are confident the conclusions also apply to the D compartment.

We find that the P compartment fully regenerates after a massive damage that kills the majority of the cells. There is a proliferative response aimed to restore the normal size of the compartment (*Figure 2*). This response is easily visualized during post-ablation time (*Figure 2E–H*), although it may start during the ablation: the fact that clones in the P compartment, which is under ablation, grow as much as those in the control A compartment (*Figure 2A–D*) suggests that they perform more cell divisions. The autonomous proliferative response of the A and P compartment provides further evidence to the proposal that A and P compartments are units of growth control (*Martin and Morata, 2006*).

One unexpected result is that the ablation of one compartment, be it the P or the D compartment, results in a transient collapse of the A/P or the D/V boundary. Cells of anterior origin can penetrate into the P compartment (*Figure 3A*) and vice versa, cells of posterior origin can penetrate into the A compartment (*Figure 3B*). The latter result is significant because the A compartment was undamaged and therefore the penetration of cells of posterior provenance cannot be due to a repair mechanism.

Possibly, the collapse of the boundary results from a transient loss (or debilitation) of the identity of cells in the P compartment (*Figure 5—figure supplement 2*) and likely some anterior cells close to the

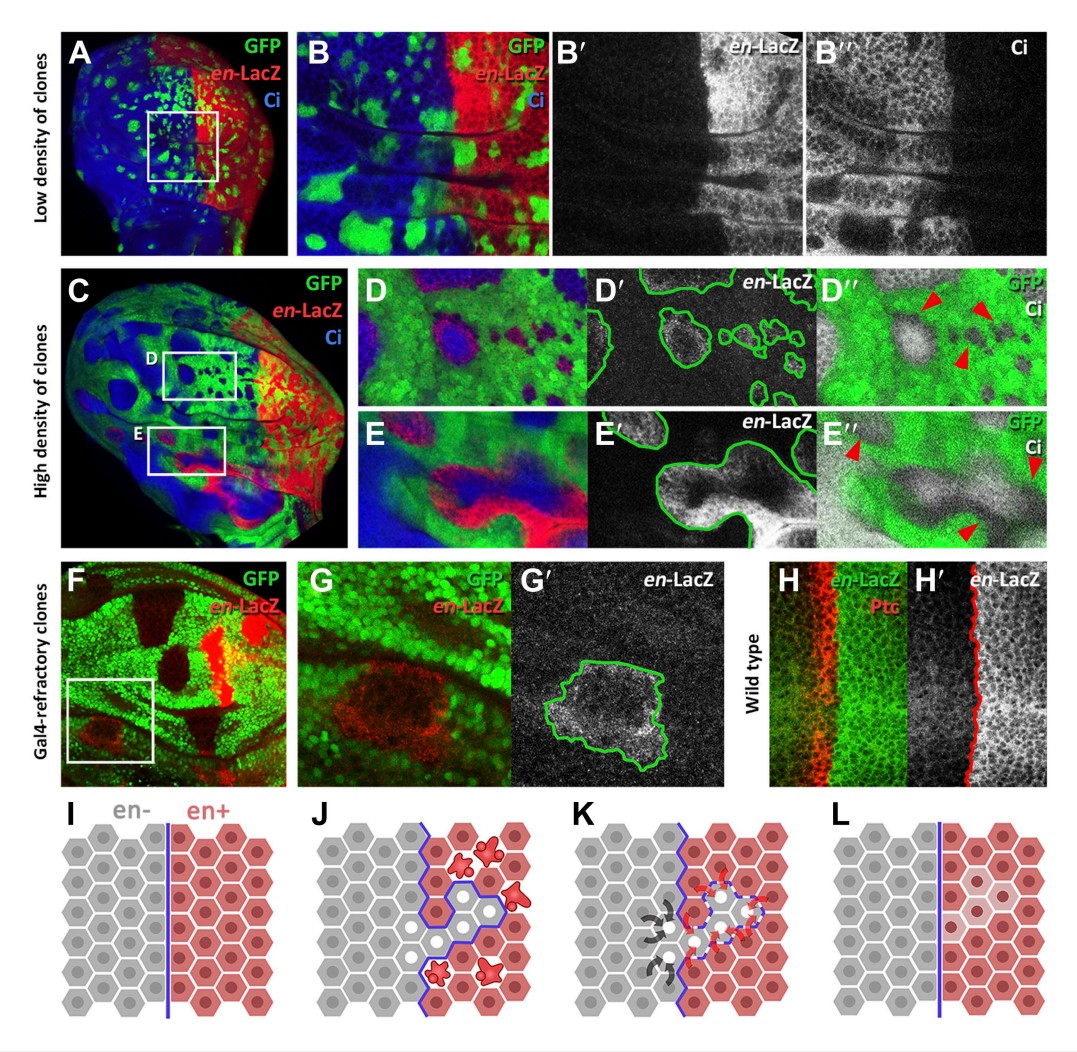

**Figure 6**. Induction of *en* activity by *en*-expressing neighbors. (**A–E'**) Clones of cells over-expressing *engrailed* and GFP were generated 48 hr before puparium formation. The clones of interest are those located in the A compartment. In A compartments with low density of *en*-expressing clones (**A**, **B–B''**) there is no induction of the *en-lacZ* reporter in anterior cells, nor there is alteration of Ci levels. However, when the disc is filled with *en*-expressing clones (**C** and magnifications in **D–E''**) anterior cells in contact with *en*-expressing clones acquire *en-LacZ* activity, while Ci protein levels decrease (**D''** and **E''**). (**F–G'**) Disc with Gal80-expressing clones (refractory to Gal4) in the anterior compartment surrounded by cells over-expressing *en*-LacZ (red) and GFP driven by the act-Gal4 line (see 'Materials and methods' for details). The clones show *en* activity, visualized by *en*-LacZ (**G**). (**H** and **H'**) High magnification of the A/P border in a control disc doubly stained for Patch, a marker of the A compartment that delineates the A/P border, and *en*-LacZ. Note that there is no extension of *en* activity beyond the boundary. (**I–L**) Scheme of our proposal of the 'induced by neighbors' model of *en* activation during the regeneration of the A/P boundary. Before cell death induction (**I**) the A and P compartments are separated by a normal straight A/P border. (**J**) During the cell killing the border collapses due to changes of identity of cells. This allows some intermingling before it is reconstructed; cells of anterior provenance may be surrounded by cells of posterior identity, which induce activity of the endogenous *en* gene of the anterior cells (**K**). Once new the identities are established, the differential affinities of the A and P cells contribute to form a new A/P boundary (**L**). See also *Figure 6—figure supplements 1 and 2*.

The following figure supplements are available for figure 6:

**Figure supplement 1**. Presence of En protein induced by *en*-expressing neighbors.

**Figure supplement 2**. Experiment of induction of clones over-expressing *en* (green) by the Gal4 system at 17°C.

border (*Figure 5—figure supplement 1*). As a consequence posterior cells can mix with juxtaposing anterior cells, in effect eliminating the boundary.

We believe the behavior of the posterior cells derives from a diminution of *en* activity, responsible for posterior identity and for maintaining the A/P border (*Morata and Lawrence, 1975*). In turn, it appears to be caused by two factors: attenuation of the control exerted by the Pc-G and trx-G genes (*Figure 5*), and up regulation of the JNK pathway (*Figure 4A*). These two pathways are known to be involved in controlling *en* activity (*Busturia and Morata, 1988*; *Breen et al., 1995*; *Gettings et al., 2010*).

We observe that the collapse of the boundary is followed by a very rapid reconstruction. The fact that there is a functional A/P border after 48 hr of ablation and without allowing time for recovery (*Figure 3—figure supplement 1A*, also *Figure 4E*), strongly suggests an immediate re-establishment. The reconstruction must result from the acquisition of new anterior or posterior identities by the cells close to the border.

We believe that the mechanism behind this reprogramming is the phenomenon of gene induction by neighbors that we report here (*Figure 6A–H'*): islands of anterior cells in which *en* is originally inactive are induced to activate the gene when surrounded by *en*-expressing cells. This phenomenon provides an explanation for the changes of identity observed in the proximity of the A/P border of regenerating discs that we outline in *Figure 6I–L*. In the developmental context of the wing disc, the cells around the A/P border only have the option of being *en-on* or *en-off*, depending on the active genotype of surrounding cells.

This non-autonomous induction may also have general implications about how major genetic/developmental decisions occur during development. In *Drosophila* many developmental decisions are taken collectively by small groups of cells: the compartmental subdivisions occur in groups of cells (*Garcia-Bellido et al., 1973*). Similarly, the phenomenon of 'transdetermination' in which cells change their original segmental determination in transplantation experiments (*Gehring, 1967*), is not a clonal but a collective decision. These examples are cases of group decisions, in which the local context plays a role. These group interactions to generate a coherent pattern of developmental resemble the 'community effect' described by John Gurdon years ago (*Gurdon, 1988*).

One could speculate that induction by neighbors may be part of a group decision process during normal development. The initial activation of *en* during embryogenesis is induced by the transcription factors *eve* and *ftz* (*DiNardo et al., 1988*). It is possible that not all the cells respond similarly to Eve or Ftz—it is known that the initial *en* expression is inhomogeneous (*DiNardo et al., 1985*). The secondary induction via the local context would ensure the coherent activation of *engrailed* in the entire group. There is a recent report suggesting the possibility that the En protein may act as a short-range signal (*Layalle et al., 2011*). It raises the possibility that the En product itself may activate *en* non-autonomously.

## Materials and methods

### *Drosophila* strains

The *Drosophila* stocks used were *hh-Gal4* (*Tanimoto et al., 2000*); *ap-Gal4* (*Calleja et al., 1996*); ET-40 (named in the text as *Psc*-QF) and QUAS-*mtdTomato* (*Potter et al., 2010*); *dpp*-LHG-86Fb (*Yagi et al., 2010*); lexO-*hid* (a kind gift of Ainhoa Pérez-Garijo and Hermann Steller); *puc-lacZ* line (pucE69) and UAS-*puc*14C line (*Martin-Blanco et al., 1998*); *act-Gal4*, UAS-*GFP*, UAS-*hid*7 (named in the text as UAS-*hid*), UAS-*Flp*, UAS-*LacZ*, tub-Gal80^TS, arm-LacZ FRT80B and *ubi-GFP* FRT80B (Bloomington Drosophila Stock Center); UAS-*en* (*Guillen et al., 1995*); hs-Flp112 and *act>stop>lacZ* (*Struhl and Basler, 1993*); ubiP63E>stop>GFP (*Evans et al., 2009*); tubP>stop>Gal80 (*Bohm et al., 2010*); mew-YFP (named in the text as PS1α-*YFP*, Kyoto Stock Center CPTI-001678); *Pc³* (*Lewis, 1978*); *trx^E2* (*Kennison and Tamkun, 1988*). In the experiments using the Gal4/UAS system the general rule was to use only two UAS transgenes. This was to avoid the possibility of titrating the amount of Gal4 protein available for the UAS vectors.

### Induction of massive cell death in the posterior and in the dorsal wing compartments

We have used a cell killing method very similar to that previously described (*Bergantinos et al., 2010*; *Smith-Bolton et al., 2009*; *Herrera et al., 2013*). For ablation of posterior compartments, we used the

genotypes UAS-*hid* *tub*-Gal80^TS/+; *hh*-Gal4/*act*>stop>*lacZ* UAS-*Flp* (experimental) and UAS-GFP *tub*-Gal80^TS/+; *hh*-Gal4/*act*>stop>*lacZ* UAS-*Flp* (control). To ablate dorsal compartment the *hh*-Gal4 insertion was substituted by an *ap*-gal4 line. The *tub*-Gal80^TS vector suppresses Gal4 activity at 17°C, but it is ineffective at 29°C thus allowing to manipulate Gal4 activity by shifting temperature between 17°C and 29°C.

Larvae were cultured at 17°C by 7 days (*Figure 1—figure supplement 1A*). At this point (approximately the transition between second and third instar), the temperature of the cultures was raised to 29°C, thus inactivating the Gal80^TS. There are two consequences of this: (1) an induction of *hid* overexpression, which induces massive apoptosis; (2) induction of Flipase recombinase, which in turn drives recombination of the *act*>stop>*lacZ* cassette, thus labeling all the cells of the Gal4 domain with indelible activity of the *LacZ* marker, which encodes the ßgal protein (*Figure 1—figure supplement 1B*). After 48 hr of ablation treatment, the cultures were returned to 17°C, activating again the Gal80^TS repressor thus allowing the recovery of the tissue. In the regenerated structure, we can distinguish the cells that belonged to the original domain (ßgal positive) from those cells originated in other compartment (ßgal negative) (*Figure 1—figure supplement 1C*).

## Inducing cell death in the Dpp domain while blocking JNK activity in the posterior compartment

We used larvae of genotypes *ubiP63E*>stop>*GFP, tub-Gal80^TS/ UAS-Flp, UAS-puc; dpp-LHG, hh-Gal4/ lexO-hid* (experimental) and *ubiP63E*>stop>*GFP, tub-Gal80^TS/ UAS-Flp; dpp-LHG, hh-Gal4/ lexO-hid* (control).

In these genotypes *hid* is overexpressed in the Dpp domain, the anterior region close to the A/P border, while *puc* (only present in the experimental genotype) and Flipase are overexpressed in the *hh* domain (posterior compartment). The Flipase in turn induces recombination of the *ubiP63E*>stop>*GFP* casette, thus labeling the posterior compartment lineage. The *tub*-Gal80^TS vector suppresses both LHG and Gal4 activity at 17°C. Larvae were cultured at 17°C for 7 days and then the temperature was raised to 29°C to inactivate Gal80^TS for 2 days. In this experiment, the transgressions were scored at the end of this ablation period.

## Measurement of transgressions

We have measured the size of transgression size as the total surface, measured in µm², of a patch of cells with a compartmental identity (activity of genes as *en* or *ci*) different from its lineage origin (e.g. Hh lineage-negative/Ci-negative or Hh lineage-positive/Ci-positive).

## Clonal analysis

To induce neutral clones in posterior compartment-ablated discs, we heat-shocked (12 min 37°C) larvae of genotype *hs-Flp; UAS-hid tub-Gal80^TS/+; hh-Gal4/ ubiP63E*>stop>*GFP*. As controls we induced clones in *hs-Flp; UAS-hid tub-Gal80^TS/+; ubiP63E*>stop>*GFP/+* larvae in which there is no Hid activation. To analyze clonal growth during the ablation period, the heat-shock was delivered at the beginning of the ablation period and the discs were fixed after 48 hr, at the end of ablation. To analyze the growth during the recovery period the heat shock was delivered at the end of Hid treatment and the discs were fixed after 72 hr.

To analyze in detail the transgression of the A/P boundary by clones generated either in the A or the P compartment we used a 'twin clones' method, marking the two cellular progenies of a mitotic recombination event. We generated the double LacZ clones and double GFP twins in the genotype *hs-Flp112; UAS-hid tub-Gal80^TS/+; ubi-GFP FRT80B hh-Gal4/arm-LacZ FRT80B*. The clones were induced by heat-shock (10 min 37°C) at the beginning of the Hid treatment. Discs were fixed after 48 hr of Hid induction and 24 additional hours of recovery period.

## Generating clones of *engrailed*-expressing cells in the anterior compartment

To generate marked clones over-expressing *engrailed* we used the construction *act*>CD2 *y*+>Gal4 (in the following *act*>stop>Gal4) (*Ito et al., 1997*). In discs of genotype *yw hs-Flp; act*>stop>Gal4 UAS-GFP/en-LacZ; UAS-en/+ a long heat shock of 30 min gave rise to a very large number of *en*-expressing clones that are labeled with GFP. For low density clones induction, we used 10 min heat shocks. We co-expressed these genes with UAS-*GFP* for labeling the clones. We detected the endogenous *engrailed* gene activity thanks to the insertion *en-LacZ* (*ryxho25*) (*Hama et al., 1990*).

## Generating clones refractory to Gal4

The genotype used was *yw hs-Flp act-Gal4 UAS-GFP; tub-Gal80^{TS}/en-LacZ; tubP>stop>Gal80/UAS-en*. In this genotype, a temperature shift from 17°C to 29°C removes the activity of the thermo-sensitive form of Gal80 thus allowing activation of the UAS-*en* transgene in all the cells of the disc (the *act*-Gal4 line confers ubiquitous expression), except in the clones resulting from recombination of the *tubP*>stop>Gal80 cassette. These clones were induced by a heat shock prior to the temperature shift and contain the normal form of the Gal80 protein. The larvae were cultured at 17°C and the heat shock (12 min at 37°C) was administrated 7 days after egg lying. After 24 additional hours at 17°C, the larvae were shifted to 29°C and the discs were fixed 2 days later.

## Immunostaining and data analysis

Immunostaining was performed as described previously (*Shlevkov and Morata, 2012*). Images were captured with Leica TCS SPE and Zeiss LSM510 Vertical confocal microscopes. Images were processed with Fiji-ImageJ and Adobe Photoshop CS4 software using Student's t for significance tests. Statistical analyses were performed with Microsoft Excel software. The following primary antibodies were used: rabbit anti-Caspase 3 (Roche, Basilea, Switzerland) 1:50; mouse anti-ß-Galactosidase (Promega, Madison, WI); rat anti-Cubitus interruptus (2A1; Hybridoma Bank, Iowa City, IA) 1:25; mouse anti-Engrailed (4D9; Hybridoma Bank) 1:50; mouse anti-Patched (Apa-1; Hybridoma Bank) 1:50; rabbit anti-H3K4me3 (07-473; Millipore, Billerica, MA) 1:50. Fluorescently labeled secondary antibodies (Molecular Probes, Eugene, OR) were used in a 1:200 dilution. TO-PRO3 (Invitrogen, Carlsbad, CA) was used in a 1:600 dilution to label nuclei; Phalloidin-Cy5 (65906; Sigma, St. Louis, MO) was used in a 1:200 dilution to label the F-actin network (thus the cell membranes).

## Acknowledgements

We thank U Banerjee, G Struhl, B Zhang, A Pérez-Garijo for fly stocks, Ernesto Sánchez-Herrero and Michael Levine for helpful discussions and comments on the manuscript.

## Additional information

### Funding

| Funder | Grant reference number | Author |
|---|---|---|
| Human Frontier Science Program | RPG0016/2010-C104 | Ginés Morata |
| Ministerio de Economia y Competitividad of Spain | BFU2008-03196 | Ginés Morata |
| JAE-Predoc fellowship - Ministerio de Economía y Competitividad of Spain | | Salvador C Herrera |

The funders had no role in study design, data collection and interpretation, or the decision to submit the work for publication.

### Author contributions

SCH, Conception and design, Acquisition of data, Analysis and interpretation of data, Drafting or revising the article; GM, Conception and design, Analysis and interpretation of data, Drafting or revising the article

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
