## [Decision Letter]

Thank you for sending your work entitled “Transgressions of compartment boundaries and cell reprogramming during regeneration in Drosophila” for consideration at *eLife*. Your article has been favorably evaluated by a Senior editor and 3 reviewers, one of whom is a member of our Board of Reviewing Editors.

The Reviewing editor and the other reviewers discussed their comments before reaching this decision, and the Reviewing editor has assembled the following comments to help you prepare a revised submission.

The three reviewers agree that this paper is potentially appropriate for *eLife* but that it does need substantial work to make it acceptable. The overriding issue that needs addressing is to make more solid and convincing the main conclusion of the paper, that there is a breakdown of compartmental boundaries that allows transgression of anterior and posterior cells. The current conclusions are interesting but somewhat superficial in places and there is a clear consensus that additional rigor is needed before it can be accepted. Specific concerns agreed by the three reviewers follow.

1) Much of the data would be interpreted differently if the b-gal genetic marker is not as precise as claimed. If 100 % of P cells are not βgal marked, then the unlabeled clones in the P compartment might not have crossed from the anterior. To exclude the possibility that the label-negative cells arise not from transgression from the healthy compartment but from a few cells that did not undergo excision of the stop cassette, the authors could mark healthy cells (in this case A cells) with the Lac-Z marker in the following genotype: dpp-LexA LexOp-Flp act>>LacZ hh-Gal4 tub-gal80ts UAS-hid. Slightly more difficult to explain, but not impossible, is the presence of marked clones in the A compartment: perhaps the dying cells induce occasional ectopic Flp expression in neighboring A cells? The controls support their model, but it is very hard to rule out that the Hid genetic manipulation might affect the normally precise control of Flp activity and/or Gal4 expression under Hh control. The twin clone experiment is harder to explain away, and perhaps is the best evidence for a breakdown of the boundary, but this is not given as much prominence as the main Hh driven βgal approach. The one example shown is at a fold, and may therefore be atypical.

2) In Figure 2, the number of clones is much too high, preventing rigorous size assessment. Is this figure representative of the pictures used to generate the data tabulated in Figure 2?

3) It appears that transgression can occur either from the disrupted to the healthy compartment (as in Figure 3) or from the healthy to the disrupted (as in Figure 3). If both phenomena are observed, what are their relative frequencies and what are the factors that affect directionality?

4) Another issue concerns the location along the boundary where transgressions occur. Transgressions are shown at various places along the boundary. An example of disrupted-to-healthy transgression is shown in the middle of the pouch (Figure 3). The twin clones are shown in the prospective notum and in a few key experiments, transgressions appear to occur preferentially at an epithelial fold. It also appears that JNK upregulation occurs specifically near folds. Transgression at these various locations seem to differ morphologically and it is not clear that they are equivalent. Therefore, it is necessary to define which types of transgression are analyzed in the different genetic backgrounds (at folds or not; towards the disrupted compartment or away from it).

5) The effect of *Pc*^*3*^*/+* on the rate of transgression into the disrupted compartment is impressive. So much so that it needs to be further substantiated (e.g., by labeling the cells of the healthy compartment as suggested above). Moreover, according to the graph (Figure 6), +/+ discs should look like an intermediate between panels A and B. This is not apparent from Figure 3.

6) In Figure 5, the authors analyze transgression size in a heterozygous background of *Pc*^*3*^*/+* or *trx*^*E2*^*/+.* They claim that the transgression size in the P compartment is altered because of changes in the epigenetic control mechanisms related to reprogramming. However, the hedgehog gene itself is under control of epigenetic regulation (Maurange and Paro, 2002), raising the possibility that the alterations in transgression size might be due to higher or lower activities of hh-GAL4 (and hence expression of hid). This could be tested by monitoring the activity of hh-Gal4 in the different genetic backgrounds using a reporter (e.g., UAS-GFP).

7) Unlike much of the rest of the data, Figure 5, are not very clear. The changes in Psc and H3K4 expression could be shown in a way that makes it easier for the reader to judge the extent of the changes.

8) In Figure 6, the authors test whether en-expressing cells can induce *en-lacZ* expression in neighboring cells. They show that this is indeed the case for cells in the anterior compartment. In the posterior compartment, where en-lacZ is normally active, however, en-lacZ is apparently repressed in clones of cells expressing Engrailed (panel B'). This raises concerns about the physiological relevance of the expression level of Engrailed in the clones and thus the validity of the conclusion. The authors could lower expression levels by using an expression system independent of the highly active Gal4/UAS system, by reducing the temperature (the Gal4 system is less active at 18C), or by co-expressing Gal80ts. Furthermore, as the authors acknowledge, engrailed normally becomes active in the A compartment at late stages of discs development. They suggest that this is not true for en-lacZ. They conclude therefore that non-autonomous engrailed expression represents a true fate change (A to P). However, the nature of the *en-lacZ* allele is not specified. This is important because at least one *en-lacz* line (known as *xho25 en-lacZ*) is upregulated in the A compartment (Blair 1992). Therefore the conclusion that engrailed is activated in a non-autonomous manner needs further evidence, perhaps with higher quality anti-En staining and a demonstration that Ptc expression is suppressed in these cells.

---

## [Author Response]

*The overriding issue that needs addressing is to make more solid and convincing the main conclusion of the paper, that there is a breakdown of compartmental boundaries that allows transgression of anterior and posterior cells. The current conclusions are interesting but somewhat superficial in places and there is a clear consensus that additional rigor is needed before it can be accepted. Specific concerns agreed by the three reviewers follow*.

We agree that this is a major point of the paper and therefore has to be demonstrated beyond doubt. We paid special attention to the various reviewers’ comments and suggestions about this issue.

*1) Much of the data would be interpreted differently if the b-gal genetic marker is not as precise as claimed. If 100 % of P cells are not βgal marked, then the unlabeled clones in the P compartment might not have crossed from the anterior*.

Although in principle it was possible that not all the P compartment cells were marked with ßgal**,** we are fully confident it is not the case for the following reasons:

A) We find that in the control discs (overexpressing GFP in the P compartment cells instead of overexpressing Hid) all the P compartment cells are labelled – no exception to this.

B) If it so happened that some P compartment cells in the experimental discs were not labelled with ßgal, the unlabelled cells should appear anywhere in the P compartment.

We have plotted the position in the disc of all the transgressions, both from A to P and from P to A and have also included the transgressions found in the *Pc*^*3*^*/+* experiment. It is obvious from the figure that the transgressions localize in the neighbourhood of the A/P border. This argues very strongly against the incomplete label hypothesis. We discuss this possibility in the revised version and are incorporating the image in the new Figure 3—figure supplement 2.

*To exclude the possibility that the label-negative cells arise not from transgression from the healthy compartment but from a few cells that did not undergo excision of the stop cassette, the authors could mark healthy cells (in this case A cells) with the Lac-Z marker in the following genotype: dpp-LexA LexOp-Flp act>>LacZ hh-Gal4 tub-gal80ts UAS-hid*.

Although the argument in the preceding paragraph argues very strongly against the possibility put forward by the reviewers, we seriously considered performing the experiment they suggested. The experiment is in principle a good one, but unfortunately it does not work. This is because the dpp-LexA line is not only expressed in the A compartment just anterior to the A/P border, but it is also expressed in the hinge region of the P compartment and in some spots elsewhere. As a consequence we are bound to find posterior cells labelled with lacZ. For the reviewers’ information we include an image of a wing disc of genotype dpp-LexA>LexO-GFP that shows GFP label in the P compartment (Figure 7).Author response image 1.

Nevertheless, you should note that in the paper we report an experiment that is a way similar to that suggested by the reviewers; in dpp-LexA/lexO-hid hh-Gal4 UAS-Flp ubi>>GFP we kill anterior cells and label healthy posterior cells marked with GFP (Figure 4). We find that posterior cells penetrate into the A compartment and express the GFP marker. In this case the expression of the dpp-LexA line in the proximal region of the P compartment should not have an effect on the results.

*Slightly more difficult to explain, but not impossible, is the presence of marked clones in the A compartment: perhaps the dying cells induce occasional ectopic Flp expression in neighboring A cells? The controls support their model, but it is very hard to rule out that the Hid genetic manipulation might affect the normally precise control of Flp activity and/or Gal4 expression under Hh control*.

The reviewers suggest the possibility that the label of anterior cells with the posterior marker lacZ (Figure 3) is not due to penetration of P cells into the A compartment, but to a transient hh expression in A cells (induced by dying posterior cells) that drives Flp and presumably also transforms them into P identity. But then these newly transformed P cells would have to change again to re-acquire anterior identity. The induction of successive changes of identity seems to us to be very unlikely.

*The twin clone experiment is harder to explain away, and perhaps is the best evidence for a breakdown of the boundary, but this is not given as much prominence as the main Hh driven βgal approach. The one example shown is at a fold, and may therefore be atypical*.

In the revised version we have modified the text to give more emphasis to these results. We selected the disc for Figure 3 because it makes a very good case for transgression. It is not frequent to have two pairs of twin clones in the same disc, both crossing the A/P border. Only one pair is close to a fold but not the other. For the reviewers’ information we are including some additional images showing other clones that are not associated with folds (Figure 8).Author response image 2.

*2) In*
Figure 2*, the number of clones is much too high, preventing rigorous size assessment. Is this figure representative of the pictures used to generate the data tabulated in*
Figure 2?

We used the disc with many clones in Figure 2 because it very clearly shows, just at a glance, the difference in size between anterior and posterior clones, but for the quantification we utilized discs with fewer clones, like those shown in the new Figure 2, that we have modified to include a typical case of a disc from the experiment

*3) It appears that transgression can occur either from the disrupted to the healthy compartment (as in*
Figure 3*) or from the healthy to the disrupted (as in*
Figure 3*). If both phenomena are observed, what are*
*their relative frequencies and what are the factors that affect directionality*?

We now include the number of cases of A to P and of P to A transgressions. The latter is clearly smaller, likely due to the fact that the number of P cells is smaller after the Hid treatment.

Concerning the factors responsible for the directionality, we discuss this issue in the Discussion section. Our argument is that there is a collapse of the A/P border due to a transient slackening of the epigenetic control exerted by the *Pc-G* and *trx-G* genes. As a result many cells of either anterior or posterior origin descend to a “naïve” identity and can mix. Due to the reprogramming during the reconstruction of the border, some cells acquire an identity different from the original one. The result is that we can observe transgressions from A to P and conversely from P to A.

*4) Another issue concerns the location along the boundary where transgressions occur. Transgressions are shown at various places along the boundary. An example of disrupted-to-healthy transgression is shown in the middle of the pouch (*Figure 3*). The twin clones are shown in the prospective notum and in a few key experiments, transgressions appear to occur preferentially at an epithelial fold. It also appears that JNK upregulation occurs specifically near folds. Transgression at these various locations seem to differ morphologically and it is not clear that they are equivalent. Therefore, it is necessary to define which types of transgression are analyzed in the different genetic backgrounds (at folds or not; towards the disrupted compartment or away from it)*.

In the new Figure 3—figure supplement 2, we show a plot of all the transgressions we have found and although there seems to be some preference for the A to P transgressions to occur close to the folding between the wing pouch and hinge, the fact is that they can occur anywhere in the disc. Curiously, the P to A transgressions show a different positional pattern. We observe upregulation of JNK in anterior cells anywhere along the A/P border – no indication of preference for folds.

*5) The effect of Pc*^*3*^*/+ on the rate of transgression into the disrupted compartment is impressive. So much so that it needs to be further substantiated (e.g., by labeling the cells of the healthy compartment as suggested above). Moreover, according to the graph (*Figure 6*), +/+ discs should look like an intermediate between panels A and B. This is not apparent from*
Figure 3.

As we mention above the experiment proposed by the reviewers would not work in practice because of the unanticipated expression of the dpp-LexA line in the P compartment.

As the reviewer says that transgressions in +/+ should be an intermediate between *Pc*^*3*^*/+* and *trx*^*E2*^*/+.*. We have modified Figure 5 to show that it is indeed the case

*6) In*
Figure 5*, the authors analyze transgression size in a heterozygous background of Pc*^*3*^*/+ or trx*^*E2*^*/+. They claim that the transgression size in the P compartment is altered because of changes in the epigenetic control mechanisms related to reprogramming. However, the hedgehog gene itself is under control of epigenetic regulation (Maurange and Paro, 2002), raising the possibility that the alterations in transgression size might be due to higher or lower activities of hh-GAL4 (and hence expression of hid). This could be tested by monitoring the activity of hh-Gal4 in the different genetic backgrounds using a reporter (e.g., UAS-GFP)*.

The reviewers raise an interesting point about possible alterations in the activity of the hh-Gal4 line (and consequently in Hid activity) in heterozygous background for *Pc*^*3*^*/+ or trx*^*E2*^*/+*. It could result in changes in the number and size of the transgressions just due to this fact.

We have monitored hh-Gal4 activity in *Pc*^*3*^*/+ and trx*^*E2*^*/+* discs by examining UAS-GFP levels – shown in the new Figure 5. We cannot detect any difference with wt controls. Also, the levels of caspase activity in the UAS-hid experiments appear to be the same for the three genotypes

*7) Unlike much of the rest of the data,*
Figure 5*, are not very clear. The changes in Psc and H3K4 expression could be shown in a way that makes it easier for the reader to judge the extent of the changes*.

In the revised version we have moved the results in the original Figure 5 to Figure 5—figure supplement 1. We present the alterations of Psc and H3K4 levels after Hid treatment in three different discs in each case. We hope the reviewer is convinced that Psc is down regulated in the vicinity of the A/P border and that the levels of H3K4 are elevated in the P compartment and also in some cells just anterior to the border

*8) In*
Figure 6*, the authors test whether en-expressing cells can induce en-lacZ expression in neighboring cells. They show that this is indeed the case for cells in the anterior compartment. In the posterior compartment, where en-lacZ is normally active, however, en-lacZ is apparently repressed in clones of cells expressing Engrailed (panel B')*.

The repression of *engrailed* in the posterior compartment by clones of *en*-expressing cells was expected. This phenomenon was reported some years ago in Garaulet et al. Development 135:3219 (2008). The *engrailed* gene, like *Ubx*, possesses a mechanism of negative auto-regulation that appears to function when the local levels are too high.

*This raises concerns about the physiological relevance of the expression level of Engrailed in the clones and thus the validity of the conclusion. The authors could lower expression levels by using an expression system independent of the highly active Gal4/UAS system, by reducing the temperature (the Gal4 system is less active at 18C), or by co-expressing Gal80ts*.

Following the suggestion by the reviewers we have repeated the experiment of *en* overexpression at 17 ºC. The result is that *engrailed* is also induced at this temperature. It is now indicated in the text and shown in the new Figure 6—figure supplement 2.

*Furthermore, as the authors acknowledge, engrailed normally becomes active in the A compartment at late stages of discs development. They suggest that this is not true for en-lacZ. They conclude therefore that non-autonomous engrailed expression represents a true fate change (A to P). However, the nature of the en-lacZ allele is not specified. This is important because at least one en-lacz line (known as xho25 en-lacZ) is upregulated in the A compartment (Blair 1992). Therefore the conclusion that engrailed is activated in a non-autonomous manner needs further evidence, perhaps with higher quality anti-En staining and a demonstration that Ptc expression is suppressed in these cells*.

We were concerned about the possibility that the *engrailed* activity in anterior cells we find might correspond to that that normally appears in late stages of the wing disc, reported by Blair. We have in many instances observed this late expression of *engrailed* using the anti-En antibody.

However, the *en-lacZ* line we have used does not show this effect. This line is indeed *ryxho25 en-lacZ*, originally reported by Hama et al. 1990, and where Blair reported *en* up regulation in the A compartment. But, in our control experiments we never find *en* expression in the A compartment. As shown in Figure 6, double staining for Ptc and lacZ in the disc indicates that *en* expression is strictly limited to posterior cells and abuts with Ptc expression in anterior cells.

We were not the only ones who failed to observe this activity – a recent paper by Landsberg et al. Current Biol **19**:1950 (2009) also failed to see it. The reviewers may examine Figure 1 of the latter paper showing a high magnification picture of a region of the wing disc; *en* expression in *ryxho25 en-lacZ* is strictly limited to posterior cells. We have re-read the 1992 Development paper by Blair. The evidence for anterior *en* expression in *ryxho25 en-lacZ* is shown in Figure 5 where one can see some local lacZ staining anterior to the A/P line. We suspect it might be an artefact generated by the staining in the peripodial membrane.

Also, as we argue in the paper, the non-autonomous induction of *engrailed* appears only when the *en*-expressing clones cover most of the A compartment (Figure 6). When there are fewer clones (Figure 6) *engrailed* is not induced. This would not be expected if the non-autonomous *en* activity were due to the Hh-dependant induction in late development reported by Blair.

In addition, we have tried to satisfy the reviewers with better anti-En staining (Figure 6—figure supplement 1) and by showing that Ptc is down regulated in anterior cells gaining *engrailed* non-autonomously (Figure 6—figure supplement 1). These results are now described in the text.